# Local drivers of Rift Valley fever outbreaks in Mauritania: A one health approach combining ecological, vector, host and livestock movement data

Yahya Barry[1☯], Markus Metz[2☯], Lina Krisztian[2], Julia Haas[2], Victoria-Leandra Brunn[2], Abdellahi Diambar Beyit[1], Ahmed El Bara[3], Ahmed Bezeid El Mamy Beyat[4], Habiboulah Habiboulah[5], Markus Neteler[2], Catherine Cêtre-Sossah[6], Elena Arsevska[6☯*]

1 Office National de Recherches et de Développement de l'Élevage et du Pastoralisme (ONARDEP), Nouakchott, Mauritania, 2 mundialis GmbH & Co. KG, Bonn, Germany, 3 Institut National de Recherches en Santé Publique (INRSP), Nouakchott, Mauritania, 4 University of Nouakchott, Nouakchott, Mauritania, 5 Direction des Services Véterinaires, Ministère de l'Élevage, Nouakchott, Mauritania, 6 French Agricultural Research for Development (CIRAD), Unit for Animals, Health, Territories, Risks and Ecosystems (UMR ASTRE), French National Institute for Agricultural Research (INRAE), Montpellier, France

☯ These authors contributed equally to this work.
* elena.arsevska@cirad.fr

**Data availability statement:** The covariates processed and used in this work are available

## Abstract

Rift Valley fever (RVF) is a vector-borne zoonotic disease with recurrent epidemic and epizootic outbreaks in Mauritania caused by the RVF virus (RVFV). In recent years, outbreaks have occurred with increasingly shorter inter-epidemic periods. The primary objective of this study was to utilise a high-resolution spatiotemporal model and identify the drivers and ecological suitability for RVFV infections, as well as areas for RVF outbreaks and emergence in humans and animals, respectively, in Mauritania. We used geolocated data from 2019 to 2023 for modelling, including human RVF cases confirmed by viral RNA detection, animal cases identified through serology or viral RNA detection, and mosquito samples in which the virus was detected by RNA analysis. Negative RVFV results were used as absence (or background) data to represent an environmental contrast between places with and without cases. Duplicates of occurrences at the exact location were kept, as multiple cases in the same place indicate a potentially higher risk. The main drivers of RVFV infection were the precipitation of the current and the preceding month of the outbreaks, followed by the average daily temperature of the current month of the outbreaks. August, September, and October were the most ecologically favourable months for RVFV infection, starting in the country's southeastern regions and expanding to the entire southern area by September and October. The RVF outbreak potential was highest in the wet season, between August and October, in most of the south and western parts of the country. Although the RVF outbreak potential is substantially reduced

at: - Total precipitation:
https://doi.org/10.5281/zenodo.12189669 -
LST: https://doi.org/10.5281/zenodo.14762335
- NDVI:
https://doi.org/10.5281/zenodo.12188840 -
NDWI:
https://doi.org/10.5281/zenodo.14764686. The
code to reproduce the current modelling results
can be found at: https://github.com/mundialis/
RVF_Mauritania/tree/1.0.0. All result figures of
potential risk maps are available at:
https://doi.org/10.5281/zenodo.14989029.

**Funding:** EA, MN, MM, LK, JH, VB received
funding from the European Commission (EC)
Grant number 874850 - H2020 MOOD project
(https://cordis.europa.eu/project/id/874850).
This work is catalogued under MOOD number
135. The views and opinions expressed in this
article are those of the authors and do not
necessarily reflect the official policy or position
of the European Commission (EC). The funders
had no role in study design, data collection and
analysis, publication decisions, or manuscript
preparation.

**Competing interests:** The authors have
declared that no competing interests exist.

during the dry season, some smaller areas in Mauritania have a relatively high outbreak potential throughout the year, and some of these areas are also located further north. These results can be used to improve sentinel active surveillance and establish an early warning model for RVFV infections in Mauritania, enabling the setting of appropriate control measures to prevent future RVF outbreaks and minimise human and animal losses.

## Author summary

Rift Valley fever (RVF) is a mosquito-borne disease that affects both animals and humans, causing serious health and economic impacts in Mauritania. Animals such as sheep, goats, cattle, and camels often suffer high death rates and abortions, while humans may experience illness ranging from mild fever to death. RVF outbreaks in Mauritania have become more frequent in recent years, highlighting the urgent need to understand when and where they are most likely to occur.

In this study, we combined data from humans, animals, and mosquitoes with climate, environmental, and livestock movement information collected between 2019 and 2023. Our modelling shows that rainfall and temperature are the main drivers of RVF outbreaks. The risk is highest in the southeastern regions at the start of the rainy season in August and spreads across the south in September and October. While the risk declines in the dry season, some areas remain vulnerable year-round.

Our results provide high-resolution risk maps for favourable environmental conditions and outbreak potential areas to guide better prevention strategies. This work demonstrates how integrating multi-sectorial data can strengthen preparedness against zoonotic diseases like RVF.

## Introduction

Rift Valley fever (RVF) is an acute viral vector-borne disease that affects ruminants and humans. In animals, the disease is characterised by high rates of abortions and perinatal death [1,2], while in humans, symptoms are generally mild; severe human cases can have haemorrhages, meningoencephalitis, retinopathy, and death [3]. Mosquitoes belonging to the *Aedes*, *Culex*, *Anopheles*, *Eretmapodites* and *Mansonia* genus are reservoirs and vectors for RVF virus (RVFV).

In Mauritania, RVF outbreaks have been reported repeatedly; the first occurred in 1987 after the construction of the Diama dam in the lower part of the Senegal River, which had ecological and environmental effects that favoured a large-scale outbreak resulting in 220 human deaths [4]. Since then, RVF epizootics/epidemics have been reported in Mauritania in 1993, 1998, 2003, 2010, 2012, 2013, 2015, 2020 and 2022 [2,5–10]. The RVF outbreaks in Mauritania occur mainly between September and November, affecting primarily sheep, goats, cattle, and camels, followed by human cases [2,5,6,8,9,11]. Animal cases often occur after grazing around temporary ponds and watering points under high mosquito pressure. Infected human patients often reported mosquito bites, contact with sick and/or aborted animals or dead animals, and nonpasteurized milk consumption [3,8].

Previous modelling studies in Mauritania [12] have demonstrated that RVF outbreaks are primarily driven by specific environmental and climatic conditions that create favourable

breeding environments for RVFV-carrying mosquitoes. These conditions include particular rainfall patterns and anomalies, as well as flooding during the rainy season, which promote the presence and rapid multiplication of competent disease vectors, leading to outbreaks notably in September and October.

More precisely, in low-level rainfall and interepidemic periods, the virus is presumably maintained by infrequent transmission from vertically infected *Aedes* genus mosquito species to susceptible intermediate hosts, such as wildlife or livestock. Periods of heavy rain leading to flooding of temporary ponds provide a suitable environment for dormant *Aedes* genus mosquito species eggs infected by RVFV to hatch and become predominant mosquito populations that transmit the virus to animals and subsequently from animals to humans. The latter primary cycle is further amplified by the proliferation of secondary vectors, such as *Culex* mosquito species populations, which, under suitable environmental conditions, can lead to large-scale RVF epizootics/epidemics that can increase with animal movements [13].

Recently, Hardcastle et al. (2020) [14] utilised RVF occurrence data from humans, animals, and vectors collected through literature review and publicly available databases from several African countries between 1995 and 2016, including Mauritania. By combining these occurrence records with ecological covariates, they predicted environmental suitability for RVF transmission at a 5 km monthly resolution across Africa. Additionally, they assessed the "spillover" potential of RVF transmission to humans and animals by incorporating population density data at the second administrative level. Across Western Africa, the highest "spillover" potential was identified in October, with the "spillover" risk persisting for between 2 and 5 months in Mauritania, particularly in the southern parts, as well as in the northern Mauritanian districts, albeit at a lower level.

Furthermore, studies of more recent RVF outbreaks, including the 2018-2019 epidemic in Mayotte [15], have shown that livestock movements play a crucial role in disease transmission. Research findings indicated that both the spread of RVFV between communes and patterns of human infection were influenced by livestock movements and proximity to areas infected with RVFV, highlighting the importance of considering the livestock movements when developing risk-based RVF surveillance and control plans.

In our current manuscript, we leverage upon the previous works from Caminade et al. (2014) [12] on the ecological drivers preceding RVF outbreaks in Mauritania, the "spillover" potential methodology proposed by Hardcastle et al. (2020) [14], and the insights on animal movement from the Mayotte RVF epidemic [15] to provide a more recent and comprehensive update of the RVF environmental suitability and outbreak potential for animals and humans in Mauritania.

We leveraged a large, multi-source dataset compiled through a collaboration between the public health and animal health sectors, comprising high-resolution, village-level data on both positive and negative occurrences of RVF in humans, animals, and mosquitoes, collected in Mauritania between 2019 and 2023. Specifically, we assessed: i) a suite of climate and environmental drivers influencing the spatial and temporal ecological suitability for RVF occurrence, and ii) the risk of RVF outbreaks in human and animal populations across Mauritania, taking into account the livestock movements and distribution of the host populations. This integrative approach enabled the development of fine-scale, spatially explicit risk maps that identify areas of high and low ecological suitability and outbreak potential at a 1 km resolution in Mauritania. These maps provide monthly and annual estimates, as well as long-term monthly risk projections, to support proactive surveillance planning and awareness raising.

## Materials and methods

### Ethics statement

This study utilised retrospective, pseudonymised data at a 1 km resolution. No personal information on patients or livestock owners was available to the authors. As this research did not involve work performed by any author on animals or humans, no ethical clearance was required.

### Data

**Disease data.** The georeferenced point data (village of residence of the patient or sampling site for the mosquitoes and animals) used in this study came from three data sources collected between 2019 and 2023 in Mauritania:

- Hospitalised, suspected and RVF-positive human cases confirmed with the reverse transcription polymerase chain reaction (RT-PCR) using the RealStar Rift Valley Fever Virus RT-PCR Kit 1.0 (Altona Diagnostics GmbH, Germany) from the epidemics of 2020 and 2022. Of 273 hospitalised patients for symptoms of hemorrhagic fever, 116 (42.5%) were positive and the remaining 157 patients were negative for RVF. The date of sampling was available for the majority of patients, except for 38 confirmed RVF cases from the 2022 epidemic for whom the date of sampling was unavailable; thus, the date of analysis was used.
- Data on animal cases originated from samples taken during the RVF outbreaks in 2020 and 2022 and the annual sentinel surveillance between 2019 and 2023 (a total of 557 sites). Of the 135 cattle sampled, 8 (5.9%) were RVFV positive with Enzyme-Linked Immunosorbent Assay (ELISA)-IgM (ID Screen Rift Valley Fever IgM Capture kit (ID.vet, Grabels, France). Of the 6600 sheep and goats sampled, 353 (5.3%) were positive for RVFV with ELISA-IgM. Of the 842 camels sampled, 210 (24.9%) were positive for RVF with RT-PCR [16]. During the RVF epidemic in 2022, three antelopes with negative RVF results with ELISA-IgM were also tested. The date of sampling was available for 555 sampling sites. For two sampling visits (small ruminant sentinel herds from 2021), the date of reception was used instead.
- Mosquito data, both positive and negative mosquito pools (5 mosquitoes per pool) confirmed with RT-PCR RVFV-specific method [16], originated from samples collected during the reinforced surveillance during the RVF outbreaks in 2020 and 2022, as well as the annual sentinel surveillance between 2019 and 2023 (from 192 sites). Of the 7693 mosquito pools, only 17 were positive (0.2%) for RVFV. The date of sampling was available for all mosquito data, except for 23 samples collected in 2021, for which the week and month were only available.

All human, animal, and mosquito georeferenced occurrences, both RVF-positive and RVF-negative results, at a given location, were combined and grouped into a monthly resolution for analysis. In general, the dating of the samples was based on the sampling date; however, if the sampling or reception date was not provided, the analysis date was used instead. Both positive and negative occurrences were used for modelling, resulting in 906 sites used as input for training the model (285 presence/positive and 621 background/negative points). In detail, the positive occurrences comprised eight mosquito, 161 animal, and 116 human sites, and the negative occurrences comprised six unknown, 132 mosquito, 326 animal, and 157 human sites. See Fig 1 for spatial distribution and Fig 2 for temporal distribution of the modelling dataset.

## Spatial distribution of disease data

**Fig 1. Spatial distribution of the RVF positive and RVF negative occurrences in Mauritania used for modelling**. Data was obtained between 2019 and 2023. The colour of the circles indicates the species of the samples. Positive occurrences: 0 unknown, 8 mosquitoes, 161 animals, 116 humans. Negative occurrences: 6 unknown, 132 mosquitoes, 326 animals, 157 humans. For details on data, see Sect 2.2.1. Mauritania's country outlines and administrative subdivisions (admin level 2) were obtained from the GADM https://gadm.org/license.html under a CC-BY open license. The dark grey map data were obtained from OpenStreetMap (admin level 4 data) https://www.openstreetmap.org/copyright.

**Environmental and climate data.** For the calculation of the ecological suitability for RVF outbreaks in Mauritania (see Sect 2.3), we used the following data:

- **Total precipitation**: monthly sum, sampled for the current month (month of sample) and the two previous months, from ERA5-Land hourly data [17] enhanced with CHELSA data [18,19] by image fusion.
- **Land Surface Temperature (LST)** Day and Night: monthly aggregates, QA filtered, cloud masked and gap-filled from MODIS LST data product MOD11A2 [20].
- **Normalised Difference Vegetation Index (NDVI)**: monthly aggregates, QA filtered, cloud masked and gap-filled from MODIS NDVI data product MOD13A2 [21].
- **Normalised Difference Water Index (NDWI)**: monthly aggregates of vegetation water content, calculated from QA filtered, cloud masked and gap-filled MODIS surface reflectance product MOD09A1 [22].

These data sets were retrieved and processed monthly for the study period from 2019 to 2023 at a spatial resolution of 30 arc seconds (approx. 1 km).

**Host and animal movement data.** For the calculation of the RVF outbreak potential in Mauritania (see Sect ), the following additional data were used:

- **Human population**: United Nations adjusted human population counts for 2020 as raster data from the WorldPop database [23].
- **Livestock population**: global distribution of cattle, sheep and goats in 2020 expressed in a total number of animals per pixel as raster data, from rasters of the Gridded Livestock of the World v4 (2020) [24,25].

## Temporal distribution of disease data

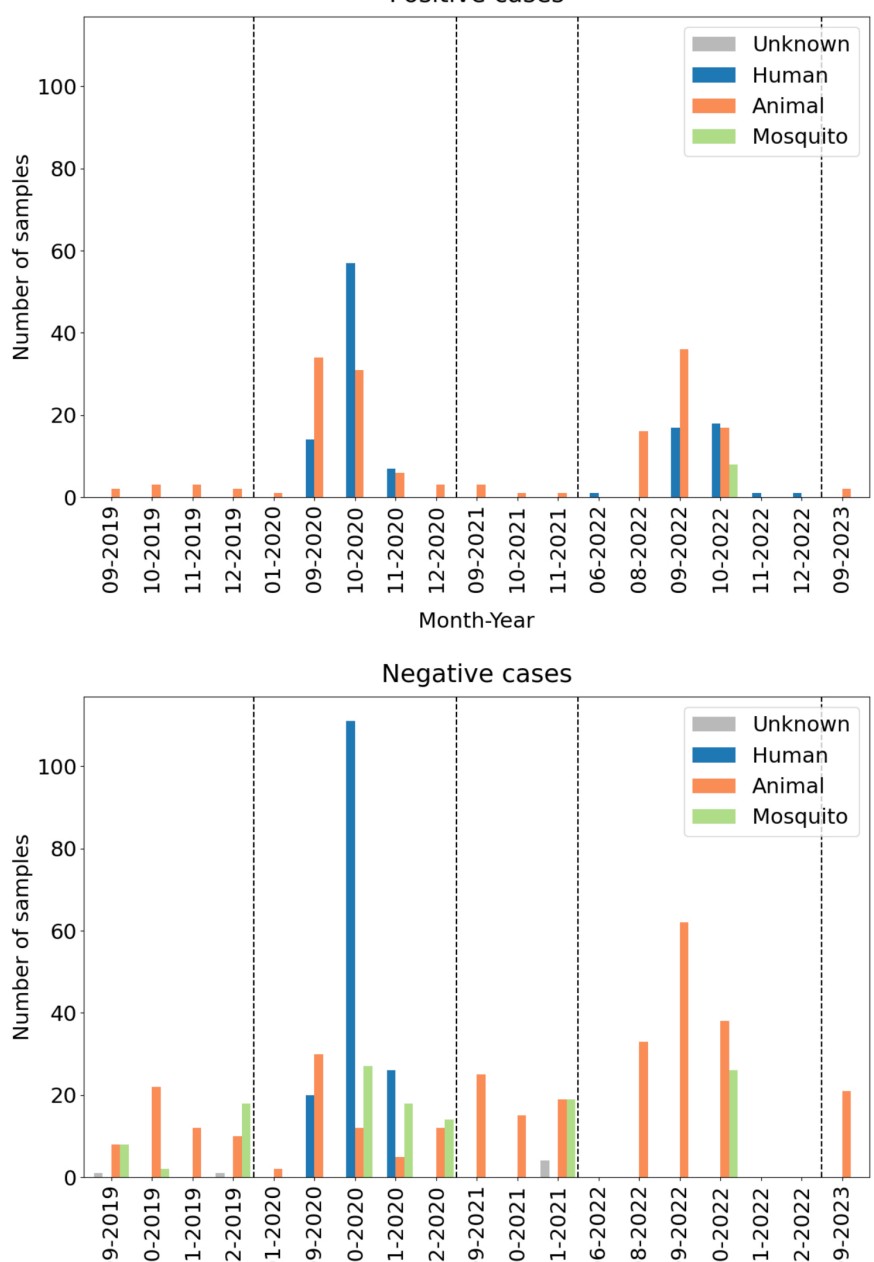

**Fig 2. Temporal distribution of the RVF positive and RVF negative occurrences in Mauritania used for modelling.** Data was obtained between 2019 and 2023. The dashed lines indicate the turn of the year. The colour of the bars indicates the species/origin of the samples. Note that the x-axis is not a continuous timeline. For details on data, see Section 2.2.1.

- **Livestock movement**: rasters of potential pathways for livestock movements in the wet and dry season, respectively, in Mauritania obtained with a landscape connectivity modelling approach [26].

These data sets were resampled to a spatial resolution of 30 arc seconds (approx. 1 km).

## Ecological suitability for RVFV infection

Disease occurrences (RVF-positive and RVF-negative) and covariate data were used in a Maximum Entropy model to predict the ecological suitability for RVFV infection in Mauritania based on similar environmental and climatic conditions (defined by covariates) [14,27,28]. The analyses were performed using a maximum entropy model at 1 km and a monthly resolution for 2019 - 2023. The open-source software *Maxent* was developed for ecological niche modelling and suitability distribution by applying a machine learning technique called maximum entropy modelling [29]. Using a set of environmental data and georeferenced occurrence localities, a probability distribution can be calculated, where each grid cell predicts the suitability of conditions for the species (here, RVFV) [29]. Specifically, the calculated probability is given by the complementary log-log (cloglog) method, which estimates the probability of presence (here, potential ecological suitability) with values between 0 and 1 [30]. For the analysis, the GRASS GIS [31] implementation of the *Maxent* tool was used.

To obtain a robust model, various model scenarios were tested: different sets of covariates were used as input, monthly models were compared to one single combined model for all occurrences, and model training was performed using all available data compared to only certain months with major RVF occurrence, duplicates (i.e., several cases at the same location) were kept vs. removed. The impact of these changes on the model was assessed using metrics such as the area under the curve (AUC) and feature importance returned by *Maxent* [32], as well as visual inspection of the resulting ecological suitability maps.

For the final model, the following covariates were selected: the precipitation of the current month, as well as of the two previous months, the LST for day and night, the NDVI and the NDWI. The disease occurrences throughout all months were used in a single model. Confirmed negative cases were used as absence (or background) data to represent an environmental contrast between places with and without cases. Duplicates of occurrences at the exact location were kept, as multiple cases in the same place indicate a potentially higher ecological suitability for RVFV infection. The model was trained with the settings described previously and performed best with an AUC of 0.68, a reasonable resulting variable contribution (not driven by only a single covariate) and potential ecological suitability maps (no geometric features/artefacts).

Subsequently, the model was applied for each month between March 2019 and December 2023, producing monthly maps of ecological suitability for RVFV infection. The first two months of 2019 were not analysed as precipitation from the preceding months was not available from the used covariate datasets. Furthermore, the mean long-term monthly predictions were calculated by averaging the monthly suitability maps over the period 2019–2023.

## RVF outbreak potential for humans and animals

The outbreak potential was defined as the ecological suitability and the presence of human and animal populations, including livestock movements, which can result in an outbreak and epidemics/epizootics [33]. The outbreak potential was determined by overlapping the monthly ecological suitability maps with human population and livestock distribution data [14,33], using the method employed by Hardcastle et al. (2020) [14]. Livestock movement

data from Jahel et al. (2020) [26] was taken into account by increasing the number of livestock per pixel in proportion to the movement of the animals through that pixel, thereby simulating increased contact between animals with increasing livestock movement.

Following the methodology proposed by Hardcastle et al. (2020) [14], a composite measure per month and geographic unit (here pixel) was calculated to keep the resolution at 30 arc seconds (approx. 1 km) provided by the data (see Sect 2.2 Data). The absolute number (population within the pixel × suitability probability per pixel) and the proportion of individuals (suitability probability of the pixel) at risk were calculated for human and livestock populations (combined for cattle, goats and sheep). Subsequently, the natural logarithm (log) was applied to those absolute and proportional measures for humans and livestock. To obtain comparable values for indicators of humans and livestock, the results were then scaled to the range of 0 to 10 based on minimum and maximum values in all dimensions (months, years and spatial units).

For the monthly outbreak potential per month and year in each geographic unit (pixel), the human population and livestock values were combined by calculating a geometric mean. Finally, the monthly outbreak potential values were combined into monthly composites, and the synoptic outbreak potential per spatial unit was determined by averaging the ranked percentiles of each month (binned into quintiles), excluding geographical units with zero outbreak potential to avoid zero-inflated histograms. For details, see S1 Appendix.

## Results

### Drivers and ecological suitability for RVFV infection

**Environmental and climate drivers.** The set of covariates chosen for the final model contributes in different shares to the model, as seen in Table 1. The percentage contribution indicates the strength of each covariate's contribution during model creation, whereas the importance of permutation specifies the contribution of each variable within the final model. The latter is determined by randomly permuting the values of the covariates among the training points (presence and background) and measuring the resulting decrease in training AUC [32].

For our model, the most important environmental variable was precipitation, especially over a longer time range indicated by the relevance of current (45%), one month prior (20%) and two months prior (14%) precipitation. Other factors involved were the LST at day (14%). The remaining covariates had a lower impact on the model with LST at night 3%, NDWI 2% and NDVI not even 1%. The relevance of the different covariates is also supported by the permutation importance values, which indicate how strongly the model depends on that variable [32].

**Table 1. Covariate contribution to the ecological suitability for RVFV infection.**

| Covariate | Percentage contribution [%] | Permutation importance [%] |
|---|---|---|
| Precipitation current month | 45.4 | 39.4 |
| Precipitation one month prior | 20.2 | 20.0 |
| Precipitation two month prior | 14.3 | 22.4 |
| LST at day | 13.6 | 8.7 |
| LST at night | 3.2 | 5.0 |
| NDWI | 2.4 | 3.1 |
| NDVI | 0.8 | 1.4 |

**Ecological suitability.** The predicted long-term monthly environmental and climate suitability for RVFV infection in Mauritania is shown in Fig 3. The predicted risk for all months between 2019 and 2023 is provided in Section Code and Data Availability.

August, September, and October were the most ecologically favourable for RVFV infection. We noted a pattern of environmental and climatic suitability that begins in the country's southeastern regions in August and expands to the entire southern area by September and October.

More precisely, in August, the most suitable ecological areas for RVFV infection are in the south of the Ouadane department, as well as in large areas of Bassiknou, Nema and Amourj departments, followed by Timbedra and Djiguenni, Aioune, Tamchakett, Boumdeid, and

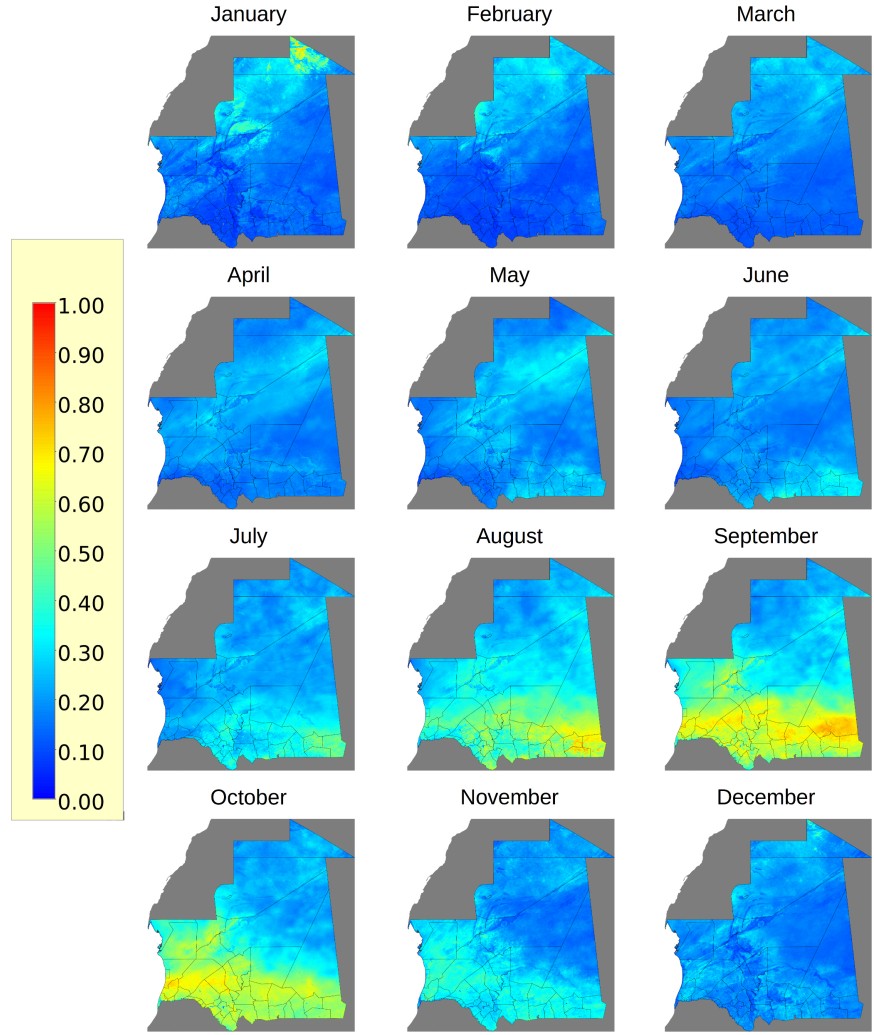

**Fig 3. Long-term predicted ecological suitability for RVFV infection**. The probability of the potential risk of RVFV infection, based on similar climatic conditions, was generated monthly by a Maximum Entropy model. For a long-term analysis of this potential risk, the monthly maps for each year were averaged, which are shown in this figure. The probability of risk is given within the range of 0 and 1, with 1 being the highest risk. Mauritania's country outlines and administrative subdivisions (admin level 2) were obtained from the GADM https://gadm.org/license.html under a CC-BY open license. The dark grey map data are obtained from OpenStreetMap (admin level 4) https://www.openstreetmap.org/copyright.

Barkeol. In September, the departments identified as having the most suitable environment and climate conditions for RVFV infection were in the south of the Ouadane department, large areas of Bassiknou, Nema, Aioune, Tamchakett, Boumdeid, Moudjeira, Maghta-Lahjar, Aleg, and Mederda. In October, higher ecologically suitable areas for RVFV infection were identified in Tamchakett, Maghtalahar, Moudjeria, Aleg, Bababe, M'Bagne, Boutmillit, Mederdra and Roso (Fig 3).

## RVF outbreak potential for humans and animals

The predicted long-term monthly RVF outbreak potential resulting in epidemics or epizootics in humans and animals in Mauritania is shown in Fig 4 and provided in Section Code and Data Availability.

The results indicate that the outbreak potential is highest during the wet season in September and October. Most of the southern and western parts of the country show high outbreak potential during the wet season. Although the outbreak potential is substantially reduced during the dry season, some smaller areas in Mauritania have a relatively high outbreak potential throughout the year, and some of these areas are also located further north. These areas are settlements with a large human and livestock population. In contrast, the northeastern part of the country, located in the Sahara Desert, has a relatively low outbreak potential during the wet season.

## Discussion

### Recommendations for monitoring and surveillance

In this study, we examined a subset of ecological factors associated with RVF outbreaks in Mauritania, enabling us to identify areas with climate and environmental conditions that may facilitate the persistence of RVFV, notably among the vector population. Additionally, by incorporating available data on host population distribution and livestock movement within Mauritania, we identified areas with a potential risk for RVF outbreaks and emergence in humans and animals.

**Monitoring precipitation and mosquitoes for early warning of potential RVFV infection**. Our findings suggest that RVFV infection in Mauritania is primarily explained by rainfall, which can be considered a surrogate for mosquito populations, as identified in previous works [11,12]. Precipitation, a key factor in producing the phytomass of the Mauritanian rangelands, creates favourable conditions for the possible circulation and persistence of the RVFV among mosquitoes. Phytomass production and, consequently, mosquito populations are highly dependent on climatic conditions, especially weekly rainfall patterns [11,12], which vary between wilayas.

More precisely, we found that the southeastern regions of the country provide suitable conditions for RVFV infection starting in July and August. This ecological suitability in the southeast can likely be attributed to the early onset of the rainy season [34,35]. The resulting waterbodies, represented by temporary and semi-permanent ponds, serve as breeding grounds and/or production sites for potential RVFV vector mosquitoes.

As winter approaches in September and October, the risk of infection expands to a broader range of departments due to the further accumulation of stagnant water that facilitates interactions between mosquito vectors and susceptible hosts [36]. The first rains in the country's southeast initiate flooding of ponds and/or water bodies, the main breeding grounds for the first generation of primary vectors belonging to the *Aedes* genus, thus enhancing the ecological suitability and initiation of the first RVFV infections [37]. Secondary vectors of RVFV,

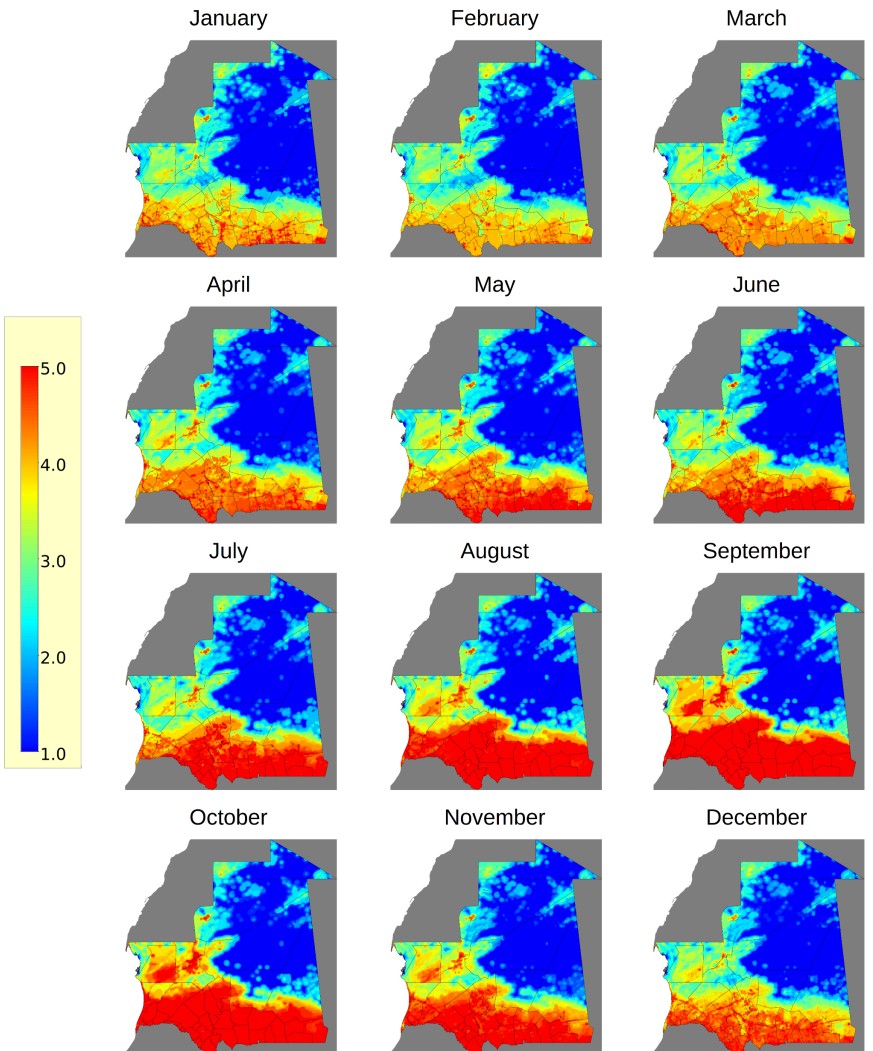

**Fig 4. Long-term predicted RVF outbreak potential (quintiles) in animals and humans in Mauritania**. The outbreak potential was determined by combining the monthly potential risk maps with human population and live-stock data. For a long-term analysis of this outbreak potential, the monthly maps for each year were combined into monthly composites, which are shown in this figure. The outbreak potential is given within the range of zero to five, with five being the highest potential. Mauritania country outlines, and administrative subdivisions (admin level 2) were obtained from the GADM https://gadm.org/license.html under an open license CC-BY. The dark grey map data are obtained from OpenStreetMap (admin level 4) https://www.openstreetmap.org/copyright.

belonging to the genus of *Culex* and *Anopheles*, will take over the transmission dynamics of RVF, leading to favourable conditions for RVFV infection in the southern parts of the country [36–38].

It is essential to note that rainfall amount and intensity can vary considerably within the same region, depending on local precipitation patterns and the distribution of stagnant water in flood-prone areas. This variation was evident during the 2010 RVF outbreaks in Adrar, where rainfall records revealed periods of cessation followed by resumption [11]. The significance of extreme rainfall events is further demonstrated by data from NASA's Soil Moisture Active Passive (SMAP) mission, as well as other soil moisture and precipitation datasets.

These showed that the 2020 Sahelian growing season, which coincided with the 2020 RVF epidemic, was the most extreme on record over the past four decades. It ranked highest for seasonally accumulated precipitation, with average rainfall exceeding long-term climatological averages by approximately 300mm [34,35].

Our model also predicted that a small area of the Bir Moghrein moughataa in the northern part of the country is ecologically suitable for RVFV infection in January, provided the virus is introduced in this area (Fig 3). This area benefits from rains in Morocco and Algeria during the Mediterranean winter season, which creates temporary water bodies. These water bodies are ideal breeding grounds for mosquitoes whilst simultaneously supporting the development of seasonal pastures. This combination increases the risk of virus transmission among dromedaries, small ruminants, and humans who congregate in the area for watering and grazing their herds [39].

**Sentinel animal surveillance for better anticipation of RVF epidemics and epizootics**. Regarding RVF emergence, our model results showed an increased risk of outbreaks during the rainy season (August-November) and, at a lower level, in the dry season (December-July), which coincides with previous studies [12]. The general pattern, with higher outbreak potential in the south and lower outbreak potential in the north, is also present in the results of Hardcastle et al. (2020) [14] for September, October, and November. However, finer spatial detail, including minor areas of high outbreak potential in the north, is missing in Hardcastle et al. (2020) [14]. In addition, the results of Hardcastle et al. (2020) [14] indicate that Mauritania has the lowest possible outbreak potential during the dry season. However, our results suggest that certain areas have a potential for outbreaks during the dry season. These results are not unexpected, as small ruminants seropositive for RVFV were detected during sentinel surveillance in December 2019 and January 2020 in Nouadhibou. Additionally, Hodh El Gharbi is also represented in our dataset (ONARDEP, author's personal communication), but is missing in Hardcastle et al. predictions (2020) [14].

Our results also showed that areas along the Senegal River in southern Mauritania are at higher risk for potential outbreaks. Given that a significant portion of the population resides in this area, it is crucial to note that it is an agricultural area of paramount importance, characterised by extensive rice and/or market gardening areas. The development of hydro-agricultural crops, including large irrigation canals, connects the river to the interior of arable land, facilitating the growth of irrigated crops. The branching river that runs through the fields throughout the year further enhances the potential for RVF outbreaks. These factors lead to residual vectors that come into contact with humans and animals, which could explain the predicted outbreak potential of RVF in these areas outside of the rainy season [40].

Our model also identified a higher outbreak potential in Nouakchott, as well as in the Ajoueft/Atar Valley, Akjoujet, and a small part of Zoueirate in F'Dérik. The outbreak potential remains high in these areas due to ecological zones favourable to the development of mosquitoes. In addition, these areas are known as livestock breeding sites, with very intense movements between the production areas of southeastern Mauritania. Because there are sites favourable to the development of RVF vectors, with the density of the human and animal population, Nouakchott has the highest human density, with the multiplication of livestock markets, where animals come from almost all the wilayas of the country. Zoueirat and F'Dérik are areas with final markets for dromedaries and small ruminants, most of which are transported by vehicle from the southeastern part of Mauritania. It is essential to take into account the transhumance movements that occur within the country during the onset of winter because the dromedaries move towards the north of the country North (Awkar) not only to escape insect bites but also to be able to perform salt cures and use the Askaf pastures of Awkar North, where there is less pressure from mosquitoes (https://dtm.iom.int/mauritania).

An outbreak risk was also observed in the northwest part of Mauritania. The north is characterised by a Saharan climate with almost no rainfall except in the case of 2010, where heavy rains were recorded in the north of the country. There are palm groves with small water areas in the oases, date palm crops, and some market gardening. Apart from RVF outbreaks in Adrar and Inchiri, livestock farming in the north is limited to transhumant small animals and camels, and traditional farming is almost non-existent (ONARDEP, personal communication).

## Perspectives and future works

Our current estimates of outbreak and emergence potential during Mauritania's dry and rainy seasons utilised data on cattle movements for commercial purposes to represent broader livestock mobility patterns [26], since cattle herders commonly move alongside small ruminants in mixed herding systems.

Future iterations of this model should incorporate species-specific movement data for camels and small ruminants. Beyond commercial movements driven by market prices, it is essential to include transhumance movement patterns, as these are primarily governed by water and pasture availability [41]. For instance, camel movements remain extensive between Mauritania's wilayas, with herds travelling as far as Mali and Senegal. Many camel herds return to northern Mauritania at the onset of winter to avoid mosquito bites and biting flies that transmit trypanosomiasis, including Tabanidae, Diptera Muscidae (house flies), and Stomoxes [42,43]. Conversely, small ruminant movements typically intensify around religious festivals, involving trade with both neighbouring and distant countries [41]. These seasonal and cultural movement patterns create distinct risk profiles that are not captured by cattle movement data alone.

Secondly, our models for ecological suitability and outbreak potential utilised monthly precipitation, temperature, NDVI, and NDWI values. We selected monthly covariates to align with RVF sentinel surveillance planning, which is conducted monthly during the rainy season, assuming that our model results would facilitate operational planning by animal health services. However, it would have been valuable to investigate whether areas at risk of infection and emergence differ when using weekly or bi-monthly spatiotemporal covariates and in particular the rainfall anomalies, as these temporal resolutions have also been demonstrated to explain past RVF outbreaks [12,13,44].

In addition, it would be valuable to examine how various climate change scenarios might influence the risk of future RVFV infection in Mauritania, as altered precipitation patterns could significantly modify disease transmission dynamics [45–47]. Climate change may also reshape animal movement patterns for both commercial and transhumance purposes, as herders adapt their farming in response to shifting availability of pastures, water resources, and market conditions [48]. Transhumance practices across the Sahel are particularly vulnerable to climate variability, with this vulnerability exacerbated by limited diversification in the primary economic sector. Recent recurrent droughts have diminished resource availability and increased health burdens for both human and animal populations [49]. These environmental pressures are already altering traditional transhumance routes, with herders increasingly moving towards northern regions of the country [50].

Therefore, future public and animal health investments must extend beyond early warning and surveillance to encompass the monitoring of socioeconomic factors and social vulnerability associated with climate change [48,51]. Collaborative efforts addressing these interconnected challenges are crucial for effective RVF preparedness and management in the face of climate change.

Finally, the RVF outbreak potential model results indicate that the risk may persist even during the dry season, albeit at substantially reduced levels compared to the rainy season. This finding raises important questions about whether low-level RVFV circulation remains undetected during dry periods, particularly given that current surveillance efforts in Mauritania are deployed exclusively during the rainy season. Future studies should investigate whether enhanced surveillance in vectors and animals during the dry season might detect low-level transmission that could serve as precursors to larger outbreaks when ecological conditions become more favourable. Further research should also characterise vector population dynamics across both rainy and dry seasons in different ecological regions, which can also be extended to inter-epidemic periods.

## Conclusion

Our modelling work demonstrates that the risk of RVF outbreaks in Mauritania is primarily driven by rainfall, with temperature being a secondary factor. The southeastern regions exhibit the highest suitability during July and August, due to early rains that create optimal breeding conditions for mosquito vectors. The risk subsequently expands southward as the rainy season progresses. At the same time, areas along the Senegal River and major peri-urban centres in the south and centre of the country face an elevated risk of RVF outbreaks due to intensive agricultural activities and livestock movements.

To effectively mitigate these risks, we recommend sentinel surveillance in both vectors and livestock as the rainy season approaches and during the rainy season, coupled with enhanced information exchange among stakeholders in identified high-risk areas. Improved case definitions should be distributed to health professionals and veterinary services, particularly at border entry points and livestock markets, with intensified screening during the first month following the onset of high rainfall.

This study builds upon previous work by providing finer spatial resolution at the pixel level, enabling the detection of smaller, high-risk areas that were previously overlooked, thereby facilitating a more targeted and efficient use of limited surveillance resources in Mauritania. Future perspectives should include capacity building in using the models and model outputs in surveillance planning, as well as updates to the risk maps, and fostering stronger collaboration between researchers, modellers, field epidemiologists, and decision-makers to ensure the effective translation of scientific findings into practical disease prevention strategies.

## Supporting information

**S1 Appendix. Outbreak potential method workflow.**
(PDF)

## Author contributions

**Conceptualization:** Markus Metz, Lina Krisztian, Julia Haas, Victoria-Leandra Brunn, Markus Neteler, Elena Arsevska.

**Data curation:** Yahya Barry, Abdellahi Diambar Beyit, Ahmed El Bara, Ahmed Bezeid El Mamy Beyat, Habiboulah Habiboulah, Elena Arsevska.

**Formal analysis:** Markus Metz, Lina Krisztian, Julia Haas, Victoria-Leandra Brunn, Markus Neteler.

**Funding acquisition:** Elena Arsevska.

**Methodology:** Markus Metz, Lina Krisztian, Julia Haas, Victoria-Leandra Brunn, Markus Neteler, Elena Arsevska.

**Supervision:** Markus Metz, Markus Neteler, Catherine Cêtre-Sossah, Elena Arsevska.

**Writing – original draft:** Yahya Barry, Markus Metz, Lina Krisztian, Julia Haas, Victoria-Leandra Brunn, Markus Neteler, Catherine Cêtre-Sossah, Elena Arsevska.

**Writing – review & editing:** Yahya Barry, Markus Metz, Lina Krisztian, Julia Haas, Abdellahi Diambar Beyit, Ahmed El Bara, Ahmed Bezeid El Mamy Beyat, Habiboulah Habiboulah, Markus Neteler, Catherine Cêtre-Sossah, Elena Arsevska.

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
