## [Decision Letter · Decision Letter 0]

6 May 2025

PNTD-D-25-00400

Local drivers of Rift Valley fever outbreaks and investigations of spillover to humans and animals in Mauritania, a One Health approach

Dear Dr. Arsevska,

Thank you for submitting your manuscript to PLOS Neglected Tropical Diseases. After careful consideration, we feel that it has merit but does not fully meet PLOS Neglected Tropical Diseases's publication criteria as it currently stands. Therefore, we invite you to submit a revised version of the manuscript that addresses the points raised during the review process.

Please submit your revised manuscript within 60 days Jul 05 2025 11:59PM. If you will need more time than this to complete your revisions, please reply to this message or contact the journal office at plosntds@plos.org. Please include the following items when submitting your revised manuscript:

We look forward to receiving your revised manuscript.

Kind regards,

Luis Adrián Diaz, Ph.D.

Academic Editor

Benn Sartorius

Section Editor

Shaden Kamhawi

co-Editor-in-Chief

Paul Brindley

co-Editor-in-Chief

**Additional Editor Comments :**

Dear Authors,

Thank you for submitting your manuscript entitled "Local drivers of Rift Valley fever outbreaks and investigations of spillover to humans and animals in Mauritania, a One Health approach" to PLOS Neglected Tropical Diseases.

We appreciate the opportunity to consider your work.

After an initial assessment and external peer review by three independent experts, we find that your manuscript addresses an important and timely topic. The proposed model for predicting Rift Valley fever outbreaks is of potential interest to the journal’s readership, particularly given the increasing attention to early warning systems for zoonotic diseases.

However, the reviewers have raised several substantive concerns that must be addressed before the manuscript can be considered for publication. The most significant issue raised across the reviews relates to the novelty and added value of your proposed model. While the work is well-structured and technically sound, the reviewers noted that it is essential to clarify and justify how your model meaningfully advances the current state of the art — whether in terms of methodological innovation, predictive accuracy, spatial or temporal resolution, or practical applicability in public health settings.

In addition, reviewers have made a number of suggestions that you can find as follows.

In light of these comments, I must request a Major Revision of your manuscript. Please address each of the reviewers' points in detail and highlight any changes made in the revised manuscript. A clear and thorough response to the issue of novelty will be critical for further consideration.

According to PLOS policies all data must be publicly available and according to one of the reviewers data used are not available.

Should you decide to submit a revised version, it will be sent back to at least one of the original reviewers for re-evaluation.

We remain interested in the topic and look forward to receiving your revised manuscript.

Sincerely,

Adrian Diaz

Academic Editor

**Journal Requirements:**

At this stage, the following Authors/Authors require contributions: Yahya Barry, Markus Metz, Lina Krisztian, Julia Haas, Victoria-Leandra Brunn, Abdellahi Diambar Beyit, Ahmed El Bara, Ahmed Bezeid El Mamy Beyat, Habiboulah Habiboulah, Markus Neteler, and Catherine Cêtre-Sossah. Please ensure that the full contributions of each author are acknowledged in the "Add/Edit/Remove Authors" section of our submission form.

4) We do not publish any copyright or trademark symbols that usually accompany proprietary names, eg ©,  ®, or TM  (e.g. next to drug or reagent names). Therefore please remove all instances of trademark/copyright symbols throughout the text, including:

- ® on page: 3.

5) Please upload all main figures as separate Figure files in .tif or .eps format. For more information about how to convert and format your figure files please see our guidelines: 

6) We noted that there are 4 figures included in the manuscript; however, there are 5 figures uploaded in the online submission form. Please check the figures labels and citations. Please also ensure that all Figure files have corresponding citations and legends within the manuscript. Currently, Figure 5 in your submission file inventory does not have an in-text citation. 

7) We noted that there is a reference to " Appendix A” on page 7; however, there is no corresponding file uploaded to the submission. Please note that the supplementary files should be uploaded as separate files with the item type 'Supporting Information'. 

8) Some material included in your submission may be copyrighted. According to PLOSu2019s copyright policy, authors who use figures or other material (e.g., graphics, clipart, maps) from another author or copyright holder must demonstrate or obtain permission to publish this material under the Creative Commons Attribution 4.0 International (CC BY 4.0) License used by PLOS journals. Please closely review the details of PLOSu2019s copyright requirements here: PLOS Licenses and Copyright. If you need to request permissions from a copyright holder, you may use PLOS's Copyright Content Permission form.

Potential Copyright Issues:

i) Figures 1, 3, and 4. Please (a) provide a direct link to the base layer of the map (i.e., the country or region border shape) and ensure this is also included in the figure legend; and (b) provide a link to the terms of use / license information for the base layer image or shapefile. We cannot publish proprietary or copyrighted maps (e.g. Google Maps, Mapquest) and the terms of use for your map base layer must be compatible with our CC BY 4.0 license.

9) Please amend your detailed Financial Disclosure statement. This is published with the article. It must therefore be completed in full sentences and contain the exact wording you wish to be published.

1) Please clarify all sources of financial support for your study. List the grants, grant numbers, and organizations that funded your study, including funding received from your institution. Please note that suppliers of material support, including research materials, should be recognized in the Acknowledgements section rather than in the Financial Disclosure

2) State the initials, alongside each funding source, of each author to receive each grant. For example: "This work was supported by the National Institutes of Health (####### to AM; ###### to CJ) and the National Science Foundation (###### to AM)."

3) State what role the funders took in the study. If the funders had no role in your study, please state: "The funders had no role in study design, data collection and analysis, decision to publish, or preparation of the manuscript."

4) If any authors received a salary from any of your funders, please state which authors and which funders.

10) Your current Financial Disclosure states, "The author(s) received no specific funding for this work.".

However, your funding information on the submission form indicates receiving a fund. Please ensure that the funders and grant numbers match between the Financial Disclosure field and the Funding Information tab in your submission form. Note that the funders must be provided in the same order in both places as well.

11) The file inventory includes files for Figures 1a, 1b, 2a and 2b. We would recommend either combining these into single Figure 1.tiff and Figure 2.tiff files with separate internal panels, or renumbering them as individual figures, as we are not able to publish multiple components of a single figure as separate files. 

**Comments to the Authors:**

**Please note that one of the reviews is uploaded as an attachment.**

**Reviewers' Comments:**

Reviewer's Responses to Questions

**Key Review Criteria Required for Acceptance?**

**Methods**

-Are the objectives of the study clearly articulated with a clear testable hypothesis stated?

-Is the study design appropriate to address the stated objectives?

-Is the population clearly described and appropriate for the hypothesis being tested?

-Is the sample size sufficient to ensure adequate power to address the hypothesis being tested?

-Were correct statistical analysis used to support conclusions?

-Are there concerns about ethical or regulatory requirements being met?

Reviewer #1: The objectives of this study are simple and well stated. I had to admit that I lack more than basic competence in the type of modeling applied in this study (but so also do many potential readers) and I think some clarification on the sources of the data used would be useful:

1. It appears that the humans and animals sampled were largely (exclusively?) from areas where RVFV infections had been formally recognized, which might makes conclusions about other areas somewhat weak.

2. Human sampling appears to have been limited to hospitalized patients (again from areas known to be undergoing RVF outbreaks) - does this not bias the sample base for the analyses that were done? Could it be that RVFV is circulating in areas that were judged to be at very low risk (northern Mauritania)? Also, as the authors describe, RVF is often very mild in humans and such patients would not likely visit a hospital.

3. 25% of the camels were RVFV positive by RT-PCR - this seems VERY high considering the transient nature of viremia for this disease - please clarify the criteria for sampling camels and what samples (sera?) were used for testing.

4. Mosquitoes: first, were they tested individually or, as is more commonly done, as pools. Second, were these mosquitoes sampled only from areas with animal or human cases?

My suggestion is to expand the description of the sampling strategy to address the questions I pose above. Also, if appropriate, add some items to the limitations section, which is generally quite good.

Reviewer #2: Pooling livestock, human, and mosquito RVFV-positive data will not inform spillover risk.

There needs to be a location-specific consideration that accounts for the expected lag between livestock and human causes.

Spillover by definition is escape of the virus from one reservoir to another -- in this case livestock to humans.

By pooling data from the two reservoirs I don't see how conclusions on spillover can be made - did I miss something?

Reviewer #3: The objectives are clearly articulated.

The study design could be improved (see attachment for my comments).

The population is clearly described and appropriate.

The sample size is sufficient.

There were no statistical analyses included, because they were not necessary.

There are no concerns about ethical or regulatory requirements.

**Results**

-Does the analysis presented match the analysis plan?

-Are the results clearly and completely presented?

-Are the figures (Tables, Images) of sufficient quality for clarity?

Reviewer #1: Results are clear and make sense; you might indicate (if you agree) that the highest value predictor (rainfall) is actually a surrogate for mosquito populations.

For the large versions of the spatial maps (end of manuscript), the text legends are quite blurry - it would be great if the text in those images could be made more clear.

Figure 2: It would seem to be much more clear if the x-axis on both positive and negative graphs were the same to make comparisons easier.

Reviewer #2: (No Response)

Reviewer #3: The analysis matched the objectives. Nevertheless, a major concern lies within the given RT-qPCR and serological data. First, the authors only mention a reference for the RT-qPCR data for the camels and for the mosquitoes. For the RT-qPCR for the human samples and for the ELISA data there is neither a reference given nor the original data shown in the manuscript. Second, while human, camel, and mosquito samples were tested by RT-qPCR, cattle, sheep, and goats were tested by ELISA. Hence, provision of the assay results and a consisting testing method would be desirable. Confirmation of ELISA data by a neutralization test would make the manuscript stronger, but is not essential.

Third, in line 82 they authors mention that “…if the sampling or reception date was not given, the analysis date was used.”. It is not clear how big the subset of samples with unknown sampling date is and when the analyses were conducted. If this subset is substantial and the analyses were conducted e.g. during August, this could strongly bias the model.

Furhermore, the model lacks disease data for Northern Mauritania. This might influence the model and should be addressed in the results or discussion part.

All Figures should be revised. More detailed figure descriptions would be helpful. Additionally, the legend fonts and axes fonts should be increased.

**Conclusions**

-Are the conclusions supported by the data presented?

-Are the limitations of analysis clearly described?

-Do the authors discuss how these data can be helpful to advance our understanding of the topic under study?

-Is public health relevance addressed?

Reviewer #1: Conclusions and limitations are good, as well as how these data may guide surveillance efforts.

Reviewer #2: (No Response)

Reviewer #3: The conclusions are supported by the presented data. Nevertheless, significant parts of discussion part can be transferred to the results section (e.g. from lines 302, 312). Please explain in more detail the characteristics of the areas that were predicted to have higher risks of spillover with regard to the following questions: Is there a higher risk for spillover in rural or urban areas? How does the risk for spillover differs in areas with high pastoralism/livestock vs. low? How big is the role of rivers and/or lakes to the risk of spillover? An addition to the manuscript would be to analyse these areas in more detail and compare them to one another. Several of these points are discussed, but could be scientifically substantiated.

It could be discussed in more detail the role of mosquitoes for spillover and outbreaks. The disease data show only very few RT-qPCR RVFV positive mosquitoes and only in October, which seems to be a discrepancy to the predictions of the spillover model, in which potential for spillover was higher from August.

Limitations could be discussed in more detail.

The authors clearly explain why the data is helpful to advance our understanding of the topic under the study and the public health relevance is addressed in detail. Nevertheless, to me the novelty of the manuscript does not become fully clear. A previous publication, which was mentioned in the manuscript as reference, shows that RVFV transmission occurs during the months of September and October in Mauritania, similarly with a modelling approach (DOI: 10.3390/ijerph110100903). In addition to this, several models for RVFV occurrence prediction have been published (e.g. DOI: 10.1073/pnas.0806490106, DOI: 10.4269/ajtmh.2010.09-0289, DOI: 10.1038/s41598-024-53774-x). Unfortunately, similarities and differences to other models were not addressed in the manuscript.

**Editorial and Data Presentation Modifications?**

Reviewer #1: Minor comments and suggestions:

Abstract: “high-resolution spatiotemporal scale” –> temporal scale model?

"molecular biology" -> virus RNA detection

2.3 section on RVF spillover potential for humans and animals: in this section, there are issues with tense: you change from describing was/were to is/are (they should be was/were)

Reviewer #2: The separation of RVF infection data into spatial and temporal, and within each separating into positive and negative, is not useful because reservoirs (i.e., livestock vs human) are pooled.

There is no information for the reader to understand how there could be a lag between livestock and human cases that would support knowledge of patters of spillover.

Reviewer #3: - Line 237: “In this study, we identified the ecological factors associated with RVF outbreaks in Mauritania…”: please formulate less strong, as not all ecological factors were considered in this model (e.g. wind).

- Line 294: “…since small ruminants positive for IgM were detected…”: better write “…since small ruminants with anti-RVFV IgM antibodies…” or “…seropositive for RVFV…”.

**Summary and General Comments**

Reviewer #1: This is a useful description of modeling to better describe risk factors for RVF in Mauritania. Overall, the methods used appear appropriate, but some clarification on the data sets used in the analyses would be valuable.

Reviewer #2: Authors are really showing RVFV transmission risk in general (to livestock and humans) and there is not enough evidence (at least in the current way it is presented and explained) for me to see fine-scaled description of spillover risk.

Reviewer #3: In general, epidemiological models are highly valuable to better understand the drivers for RVF outbreaks and spillover events. The topic is highly relevant for public health interventions and informed surveillance strategies as well as intervention strategies. However, I am not convinced about the novelty of this study as well as about the presentation of the data.

The manuscript describes a model for investigations of ecological drivers and spillover of Rift Valley fever virus (RVFV) in Mauritania. The authors use, compared to previous publications, recent data from samples collected between 2019 and 2023 in Mauritania. It seems like the authors main novelty is the use of recent data. The model which was established newly, did not come to any different conclusions than other models used before. The main results, that the wet season is a main driver of RVF occurrence and spillover in Mauritania, were seen before with other models.

Additionally, the raw data for the virological assays are missing. These data should be made publicly available.

To improve the manuscript, e.g. the influence of human - livestock interaction (consumption of meat, unpasteurized milk, husbandry) could be further explored. In addition, the role of mosquitoes should be assessed in more detail. Besides this influence of rural vs urban areas, proximity to lakes or rivers could be included in the model and assessed in more detail (for further explanations, please see my attached document).

PLOS authors have the option to publish the peer review history of their article (what does this mean?). If published, this will include your full peer review and any attached files.

Reviewer #1: No

Reviewer #2: No

Reviewer #3: No

**Figure resubmission:**
---

## [Decision Letter · Decision Letter 1]

10 Sep 2025

Dear Dr Arsevska,

We are pleased to inform you that your manuscript 'Local drivers of Rift Valley fever outbreaks in Mauritania: A One Health approach combining ecological, vector, host and livestock movement data' has been provisionally accepted for publication in PLOS Neglected Tropical Diseases.

Best regards,

Luis Adrián Diaz, Ph.D.

Academic Editor

Benn Sartorius

Section Editor

Shaden Kamhawi

co-Editor-in-Chief

Paul Brindley

co-Editor-in-Chief

Reviewer's Responses to Questions

**Key Review Criteria Required for Acceptance?**

**Methods**

-Are the objectives of the study clearly articulated with a clear testable hypothesis stated?

-Is the study design appropriate to address the stated objectives?

-Is the population clearly described and appropriate for the hypothesis being tested?

-Is the sample size sufficient to ensure adequate power to address the hypothesis being tested?

-Were correct statistical analysis used to support conclusions?

-Are there concerns about ethical or regulatory requirements being met?

Reviewer #3: The objectives of this study are clearly stated and the study design is appropriate in my opinion. The authors addressed my concerns regarding the RT-qPCR and ELISA data sufficiently. The population is now well described. There are no ethical or regulatory concerns.

**Results**

-Does the analysis presented match the analysis plan?

-Are the results clearly and completely presented?

-Are the figures (Tables, Images) of sufficient quality for clarity?

Reviewer #3: The presented analysis matched their objectives and the results are clearly and completely presented now. The figures have been adapted to be visually more attractive (increased font sizes).

**Conclusions**

-Are the conclusions supported by the data presented?

-Are the limitations of analysis clearly described?

-Do the authors discuss how these data can be helpful to advance our understanding of the topic under study?

-Is public health relevance addressed?

Reviewer #3: The conclusion is supported by the presented data and the limitations are stated. There is a good discussion on how this data could inform politicians or other stakeholders. The public health relevance is clearly addressed and suggestions to improve surveillance strategies are given. My questions regarding the conclusion were answered.

**Editorial and Data Presentation Modifications?**

Reviewer #3: (No Response)

**Summary and General Comments**

Reviewer #3: I would like to thank the authors for their good clarification and adaption in the manuscript on the novelty and added value of this study, which now is clearer to me.

PLOS authors have the option to publish the peer review history of their article (what does this mean?). If published, this will include your full peer review and any attached files.

Reviewer #3: No

---

## [Editor Report · Acceptance letter]

Dear Dr Arsevska,

We are delighted to inform you that your manuscript, "Local drivers of Rift Valley fever outbreaks in Mauritania: A One Health approach combining ecological, vector, host and livestock movement data," has been formally accepted for publication in PLOS Neglected Tropical Diseases.

Best regards,

Shaden Kamhawi

co-Editor-in-Chief

Paul Brindley

co-Editor-in-Chief
